# Antiplasmodial Activity of *Vachellia xanthophloea* (Benth.) P.J.H. Hurter (African Fever Tree) and Its Constituents

**DOI:** 10.3390/ph15040470

**Published:** 2022-04-13

**Authors:** Nasir Tajuddeen, Tarryn Swart, Heinrich C. Hoppe, Fanie R. van Heerden

**Affiliations:** 1School of Chemistry and Physics, University of KwaZulu-Natal, Private Bag X01, Scottsville 3209, South Africa; ntajuddeen@yahoo.com; 2Department of Biochemistry & Microbiology, Rhodes University, Grahamstown 6140, South Africa; g10s2905@campus.ru.ac.za (T.S.); h.hoppe@ru.ac.za (H.C.H.)

**Keywords:** *Vachellia xanthophloea*, Fabaceae, flavonoids, methyl gallate, malaria, *Plasmodium*

## Abstract

*Vachellia xanthophloea* is used in Zulu traditional medicine as an antimalarial remedy. A moderate antiplasmodial activity was previously reported for extracts of the plant against D10 *Plasmodium falciparum*. This study aimed to identify the phytochemicals responsible for the antiplasmodial activity of the leaf extract. The compounds were isolated by chromatography and their structures were determined using spectroscopic and spectrometric methods. The antiplasmodial activity was evaluated using a parasite lactate dehydrogenase assay and cytotoxicity was determined using a resazurin assay. The ethyl acetate fraction inhibited *P*. *falciparum* with IC_50_ = 10.6 µg/mL and showed minimal cytotoxicity (98% cell viability at 33 µg/mL). The chromatographic purification of this fraction afforded sixteen compounds, including two new flavonoids. A 1:1 mixture of phytol and lupeol was also isolated from the hexane fraction. All the compounds were reported from *V. xanthophloea* for the first time. Among the isolated metabolites, methyl gallate displayed the best activity against *P. falciparum* (IC_50_ = 1.2 µg/mL), with a 68% viability of HeLa cells at 10 µg/mL. Therefore, methyl gallate was responsible for the antiplasmodial activity of the *V. xanthophloea* leaf extract and its presence in the leaf extract might account for the folkloric use of the plant as an antimalarial remedy.

## 1. Introduction

*Acacia* (Fabaceae) is a genus of diverse trees and shrubs widely distributed in Africa, Australia, Asia, and the Americas. The genus was reclassified in 2003 at the 17th International Botanical Congress after the approval of a proposal by an Australian group of botanists. The name *Acacia* was retained for the Australian species, whereas the African species were grouped into two genera, *Vachellia* and *Senegalia*, based on their morphological, anatomical, and biochemical characteristics [1,2]. Thus, *Acacia xanthophloea* was renamed *V. xanthophloea* based on this reclassification. *V. xanthophloea* (Benth.) P.J.H. Hurter (Fabaceae) (syn. *Acacia xanthophloea*) is known in isiZulu as *umkhanyagude* (to be bright or to shine from afar) [3] apparently due to the iconic bright yellow-green colouration of its stem bark, which appears to be powdery in several locations. *V. xanthophloea* has long been associated with malaria. Before the cause of malaria was understood, early discoverers thought that the tree caused malaria because people living or travelling in areas where the trees grew usually found themselves with a fever and thus named it the ‘fever tree’ [3,4]. However, this is only a coincidence as the tree mainly grows in swampy areas, an ideal breeding ground for mosquitoes, the vector for the malaria parasite. In Zulu traditional medicine, an emetic prepared from the powdered stem bark and roots is used for malaria treatment and prevention. In Tanzania, the powdered stem bark and root bark are taken as a remedy for malaria and root and stem bark decoctions are used against abdominal pains and anaemia, respectively. The Luvale women in Angola and Zambia use cold root infusions from a tree purported to be *V. xanthophloea* as a vaginal wash for the relief of abdominal pain [5,6]. *V. xanthophloea* is listed among the traditional fever remedies in Zambian folkloric medicine, and most fevers in Zambia are due to malaria [7]. Interviews conducted with traditional health practitioners in Meru County at the Imenti forest game reserve, in Tharaka Nithi County at Gatunga in Kenya [8], and in the KwaZulu-Natal province in South Africa [9] indicated that *V. xanthophloea* is used to treat malaria. Furthermore, information from traditional healers showed that the plant is used by people in South Africa to treat the symptoms of tuberculosis, such as a fever, a cough, and blood in the sputum [10]. The Masai people in the Loitokitok district of Kenya use it against skin disorders and to relieve fatigue [11]. A decoction made from *V. xanthophloea* stem bark and four other medicinal plants is used against infertility by women in Kenya, and the Zulus in South Africa prepare a love charm by mixing the plant with *Lithops lesliei* N.E.Br. (Aizoaceae) [12]. Previous biological studies have shown that the stem bark extract of *V. xanthophloea* inhibited the D10 strain of *P. falciparum* [13] but no compounds were isolated. As part of a project aimed at discovering the antiplasmodial compounds from South African ethnomedicinal plants [14,15,16], we report in this paper the antiplasmodial and cytotoxic activities of *V. xanthophloea* leaf extract and its isolated compounds. To the best of our knowledge, this is the first report on the phytochemistry of *V. xanthophloea*.

## 2. Results and Discussion

### 2.1. Chemistry

The ethyl acetate fraction of *V. xanthophloea* leaf extract displayed good antiplasmodial activity (IC_50_ = 10.6 µg/mL) and was not cytotoxic (98% cell viability at 33 µg/mL). Therefore, the extract was subjected to various chromatographic purification techniques to afford sixteen compounds. The structures of the compounds were elucidated by analyses of the nuclear magnetic resonance (NMR) and mass spectral data and by comparison with the data in the literature. The structures of the isolated compounds are presented in Figure 1.

Compounds **1** and **2** were obtained as an inseparable mixture (approximately 1:1 based on NMR integration) as a pale yellow amorphous solid. Careful and extensive analyses of the NMR spectra (Appendix A) allowed the structural elucidation of the individual compounds. Compound **1** was assigned a molecular formula of C_21_H_22_O_8_ based on the pseudo-molecular ion [M + H]^+^ at *m/z* 403.1401 (calculated for C_21_H_23_O_8_ 403.1393) observed in the high-resolution electrospray ionisation mass spectrum (HR-ESI-(+)-MS) (Appendix A), suggesting eleven degrees of unsaturation. The ^1^H NMR spectrum of compound **1** revealed a pair of *ortho*-coupled aromatic doublets at δ_H_ 8.00 (1H, d, *J* = 9.0 Hz, H-5) and δ_H_ 7.03 (1H, d, *J* = 9.0 Hz, H-6). These ^1^H NMR data (Table 1) were typical of H-5 and H-6 of 5-deoxyflavones with an oxygen function at position 7, and fell within the ranges of the chemical shifts of H-5 (7.9–8.2 ppm) and H-6 (6.7–7.1 ppm) [17]. In addition, 2 aromatic singlets at δ_H_ 7.32 (1H, s, H-6′) and δ_H_ 6.64 (1H, s, H-3′) and 6 aromatic methoxy singlets at δ_H_ 3.98 (3H, s, OCH_3_), 3.97 (3H, s, OCH_3_), 3.95 (3H, s, OCH_3_), 3.86 (3H, s, OCH_3_), 3.85 (3H, s, OCH_3_), and 3.80 (3H, s, OCH_3_) were observed in the ^1^H NMR spectrum. The ^13^C and distortionless enhancement by polarisation transfer (DEPT) NMR spectra of compound **1** showed 14 aromatic signals (Table 1), of which 8 were oxygenated with 6 methoxy carbons at δ_C_ 56.3–61.5 and a carbonyl carbon at δ_C_ 173.4. All protonated carbons were assigned based on heteronuclear single quantum coherence (HSQC) cross-peaks. These ^13^C NMR data suggested that Compound **1** was a polymethoxylated flavonoid; this was in agreement with the assigned molecular formula C_21_H_22_O_8_.

The appearance of the two aromatic signals at δ_H_ 7.32 and δ_H_ 6.64 as singlets suggested that the two protons were not in close proximity of each other and could be assigned to two *para*-substituted protons on ring B. Alternatively, one of the signals could be assigned to C-3 of a flavonoid and the other signal placed on ring B. The absence of a correlation in the HMBC spectrum between any of the singlet protons and the carbonyl at C-4 supported the assignment of two protons on aromatic ring B. The assignment of the two singlets at positions H-3′ and H-6′ was further supported by the HMBC correlations between H-3′ and H-6′ with C-1′ and that of only H-6′ with C-2. The placement of the two protons resonating as doublets at δ_H_ 8.00 and δ_H_ 7.03 in the *ortho*-positions at H-5 and H-6 on ring A was supported by the COSY correlation between H-5 and H-6. The assignment of H-5 and H-6 was confirmed by the long-range HMBC correlations of H-5 with C-4 and C-9 as well as that of H-6 with C-10. Five of the methoxy groups were located at C-7, C-8, C-2′, C-4′, and C5′ based on the HMBC correlations of δ_H_ 3.98 with 156.1 (C-7), δ_H_ 3.95 with 136.8 (C-8), δ_H_ 3.97 with 151.9 (C-2′), δ_H_ 3.85 with 152.7 (C-4′), and δ_H_ 3.86 with 143.0 (C-5′). The sixth methoxy group was placed at C-3 based on the downfield chemical shift of C-3 (δ_C_ 141.3) and the HMBC correlation between δ_H_ 3.80 and 141.3. The above NMR data indicated that Compound **1** was a 5-deoxyflavonoid with a 3,7,8,2′,4′,5′-hexasubstitution pattern. The structure was assigned as 3,7,8,2′,4′,5′-hexamethoxyflavone 1 (Figure 1), a new natural product.

Compound **2** was isolated as a mixture with compound **1**, and the HR-ESI-(+)-MS showed a pseudo-molecular ion [M + H]^+^ at *m/z* 389.1248 (calculated for C_20_H_21_O_8_ 389.1236), suggesting a molecular formula of C_20_H_20_O_8_. The ^1^H and ^13^C NMR spectra of compound **2** were similar to those of compound **1**. However, compound **2** had one less methoxy signal compared with compound **1** in addition to the appearance of a proton singlet at δ_H_ 8.15 (1H, s, OH). An analysis of the HSQC spectrum of compound **2** showed that this proton was not attached to any carbon atom, indicating that it was a hydroxy proton. An examination of the HMBC spectrum showed a cross-peak between the hydroxy proton and δ_C_ 151.3 (C-2′), suggesting that the *O*-methyl group at C-2′ of compound **1** had been replaced by OH. A comparison of the NMR data of compound **2** with the literature data [18,19] showed that the compound was 2′-hydroxy-3,7,8,4′5′-pentamethoxyflavone **2** (Figure 1). The compound was previously only reported for *Mimosa diplotricha* and *Parkia clappertoniana* [18,19], both part of the Fabaceae.

Compound **3** was isolated as a yellow solid and a molecular formula of C_18_H_16_O_8_ was assigned based on a pseudo-molecular ion [M − H]^−^ at *m/z* 359.0776 (calculated for C_18_H_15_O_8_ 359.0767) observed in the HR-ESI-(-)-MS (Appendix A). The ^1^H NMR spectrum (Appendix A) showed signals for a pair of *meta*-coupled aromatic doublets at δ_H_ 6.33 (1H, d, *J* = 2.0 Hz, H-8) and δ_H_ 6.20 (1H, d, *J* = 2.0 Hz, H-6), assignable to a flavonoid phloroglucinol ring A, two aromatic singlets at δ_H_ 7.02 (1H, s, H-3′) and δ_H_ 6.60 (1H, s, H-6′), and three methoxy singlets at δ_H_ 3.88 (3H, s, OCH_3_), δ_H_ 3.83 (3H, s, OCH_3_), and δ_H_ 3.73 (3H, s, OCH_3_) (Table 1). The COSY spectrum (Appendix A) showed the expected correlation between δ_H_ 6.20 (H-6) and δ_H_ 6.33 (H-8). The ^13^C NMR spectrum showed a total of 18 resonances (Table 1) between δ_C_ 180.3 and 55.4, including a carbonyl at δ_C_ 180.3, 8 oxygenated aromatic carbons between δ_C_ 166.2 and δ_C_ 139.4, and 3 methoxy carbons between δ_C_ 60.7 and δ_C_ 55.4, suggesting a trimethoxylated flavonoid nucleus. All the protonated carbons were assigned based on the HSQC cross-peaks. The HMBC spectrum (Appendix A) did not show a correlation between any of the two aromatic singlets in the ^1^H NMR spectrum and the carbonyl carbon, thereby precluding a proton on C-3. It followed that both singlets must be assigned to ring B at *para*-positions relative to each other. The observed HMBC correlations between δ_H_ 7.02 (H-3′) and δ_H_ 6.60 (H-6′) with δ_C_ 153.4 (C-4′), δ_C_ 151.0 (C-2′), δ_C_ 142.5 (C-5′), and δ_C_ 156.6 (C-2) confirmed the assignment. Two methoxy groups were placed at C-4′ and C-5′ based on the HMBC correlations between the singlet at δ_H_ 3.88 and δ_C_ 153.4 (C-4′) and between the singlet at δ_H_ 3.83 and δ_C_ 142.5 (C-5′). The third methoxy group was assigned to C-3 based on the downfield chemical shift of C-3 (δ_C_ 138.6) and the HMBC correlation of the singlet at δ_H_ 3.73 with δ_C_ 138.6. Based on the above analysis, the structure of the new compound **3** was assigned as 5,7,2′-trihydroxy-3,4′,5′-trimethoxyflavone (Figure 1).

The known compounds were identified as 3-*O*-methylquercetin (**4**) [20], quercetin (**5**) [21], dihydroquercetin (**6**) [22], catechin (**7**) [23], gallocatechin (**8**) [23], methyl gallate (**9**) [24], kaempferol (**10**) [25], apigenin (**11**) [26], pinoresinol (**12**) [27], *E*-lutein (**13**) [28], 1-heptacosanol (**14**) [29], phytol (**15**) [30], and lupeol (**16**) [16].

Most of the compounds isolated from the leaves of *V. xanthophloea* in this study were flavonoids and were reported from the plant for the first time. A few of the flavonoids and their glycosides have been previously isolated from other *Vachellia* species (syn. Acacia). Apigenin and quercetin were reported from the leaves of *A. tortilis* (now *Vachellia tortilis* (Forssk.) Gallaso & Banfi) [31]. Glycosides of apigenin, quercetin, and kaempferol were isolated from the leaves of *Acacia pennata* (now *Senegalia pennata* (L.) Maslin) [32]. However, to the best of our knowledge, this is the first report of polymethoxylated flavonoids from a *Vachellia* species. The isolation of methyl gallate from *V. xanthophloea* was consistent with the phytochemistry of the genus *Vachellia*. Several phenolic acid derivatives have been isolated from the species of *Vachellia*. Methyl gallate was reported in *A. nilotica* (now *Vachellia nilotica* (L.) P.J.H. Hurter & Mabb. subsp. *kraussiana* (Benth) Kyal. & Boatwr.) [33] and *A. farnesiana* (now *Vachellia farnesiana* (L.) Wight & Arn.) [34]. Ethyl gallate was also isolated from *A. nilotica* (now *V. nilotica* (L.) P.J.H. Hurter & Mabb.) [33], and benzoic acid derivatives were isolated from *A. confusa* Merr. [35]. Lignans are not commonly found in *Vachellia* and *Senegalia* genera [36]. The isolation of pinoresinol from *V. xanthophloea* is the first report of a lignan from any *Vachellia* or *Senegalia* species.

### 2.2. Biological Activity

The antiplasmodial and cytotoxic activities of the extract and a few of the isolated compounds against the chloroquine-sensitive 3D7 clone of *P*. *falciparum* and HeLa cells are summarised in Table 2 and Table 3. The ethyl acetate extract showed a good antiplasmodial activity and was not cytotoxic (98% cell viability at 33 µg/mL). Among the isolated compounds, methyl gallate (**9**) showed the best antiparasitic activity (IC_50_ 1.2 µg/mL). Although it was cytotoxic against HeLa cells at 50 µg/mL (2.0% HeLa cell viability), the cytotoxicity was reduced at 10 µg/mL (68% viability of HeLa cells), suggesting a degree of selectivity for 3D7 parasites. The flavonoids inhibited the parasite viability at a high concentration of 50 µg/mL but the compounds were equally cytotoxic. The antiplasmodial activity and cytotoxicity were lost at a lower concentration of 10 µg/mL. Considering that the leaf extract was not cytotoxic, it is possible that the coexistence of all the compounds in the mixture acted in ways that mitigated the cytotoxicity. It is worthy to note that when a mixture of methyl gallate (**9**) and 3-*O*-methylquercetin (**4**) (with 82.9% parasite viability at 10 µg/mL) was tested, the antiplasmodial activity was lower than that of methyl gallate alone (Table 2). Pinoresinol (**12**) and gallocatechin (**8**) have previously been reported to be inactive against *P*. *falciparum* [37,38].

It has been reported by other investigators that methyl gallate shows good selective antiplasmodial activity [39]. The compound presented a good pharmacokinetic profile and satisfied the Lipinski rule for drug-likeness in an in silico prediction. It was also shown to act in synergy with quinine as well as on late-stage parasite schizonts and trophozoites [40]. Common dietary flavonoids such as kaempferol, apigenin, quercetin, myricetin, lutenin, and glycosides have been reported to possess a moderate to weak antiplasmodial activity [41,42]. In this study, most of the flavonoids were also cytotoxic to HeLa cells at the concentration in which the activity against the malaria parasite was significant, suggesting that the antiplasmodial activity might be due to general cytotoxicity.

As the most active among the isolated metabolites was methyl gallate, the question arose as to whether the compound was an artefact formed by transesterification with the methanol used in the extraction process. A high-performance liquid chromatography (HPLC) analysis of a freshly prepared extract—obtained under mild conditions by extracting the fresh plant material with an extraction solvent for 1 h at room temperature and away from a light source—showed that methyl gallate was present, thus confirming that it was a true natural product in *V*. *xanthophloea*. The majority of the compounds isolated from the leaf extract of *V*. *xanthophloea* were flavonoids and they showed a moderate to weak antiplasmodial activity. There are conflicting accounts of the reported antiplasmodial activity and selectivity of flavonoids against different strains of *P*. *falciparum*. Reports of potent and selective to weak and unselective flavonoids against *P*. *falciparum* can be found in the literature. Antiplasmodial activity has been reported for prenylated, methoxylated, glycosylated, and common dietary flavonoids [41,42]. The discrepancy in the antiplasmodial activity profile of flavonoids might be due to the different assay methods and different *P*. *falciparum* strains. Nevertheless, the further development of flavonoids as viable antiplasmodial agents has been hampered by the apparent lack of selectivity in their bioactivity. Considering the abundance of flavonoids in dietary sources and how easily the flavonoids could be harnessed to combat malaria if the efficacy is proven, a detailed structure/activity relationship study on the antimalarial activity of flavonoids is worthwhile. It has been argued that the presence of flavonoids in extracts might play a secondary role in the activity of the extract. In this regard, the flavonoids could act by potentiating the activity of the active compound in the extract through additivity or synergism. For example, the infusion of *Artemisia annua* L. is used in traditional Chinese medicine (TCM) to treat malaria and contains flavonoids, several of which are inactive against the malaria parasite alone but have been shown to potentiate the activity of artemisinin in a combination [43]. The *A*. *annua* infusion used in TCM contains only about 20% of the recommended daily dosage of artemisinin used in conventional treatments [43]. The role of the flavonoids isolated from the leaf extract of *V*. *xanthophloea* in the antiplasmodial activity of the extract was not clear and should be further investigated.

## 3. Materials and Methods

### 3.1. General Procedures

Optical rotations were measured on a Bellingham and Stanley ADP440+ polarimeter. ^1^H NMR (400 MHz, 500 MHz) and ^13^C NMR (100 MHz, 125 MHz for) spectra were recorded on Bruker AVANCE III spectrometers using a 5 mm BBOZ probe with CD_3_OD (δ_H_ 3.31, δ_C_ 49.03), CDCl_3_ (δ_H_ 7.26, δ_C_ 77.06), and DMSO-*d_6_* (δ_H_ 2.50, δ_C_ 39.53) as solvents. The residual solvent peaks were used as internal standards. High-resolution mass spectra were obtained on a time-of-flight (TOF) mass spectrometer (Waters Micromass LCT Premier) in MS-grade acetonitrile and methanol solutions. The ionisation source was an ESI in the negative or positive mode. The GC-MS analysis was performed on a GC-MS-QP2010SE (Shimadzu) gas chromatograph-mass spectrometer. Pre-coated TLC silica gel 60 F_254_ (Merck) plates were used for the thin-layer chromatographic analyses. The column chromatography was performed on a Merck silica gel (230–400) mesh and Sephadex LH-20 (Fluka) was used for the gel filtration chromatography. UV light at 254 and 365 nm was used on the viewing spots on the TLC plates before a visualisation by spraying with a *p*-anisaldehyde/H_2_SO_4_ spray reagent (0.5 mL *p*-anisaldehyde, 10 mL glacial acetic acid, 4 mL concentrated H_2_SO_4_ acid, and 85 mL MeOH) and heating at 100 °C for 5 min. The hexanes used for the column chromatography referred to an isomeric mixture of hexanes separated at boiling point and commercially referred to as hexane.

The HPLC experiments were performed on a Shimadzu LC-20AB Prominence liquid chromatograph equipped with a binary pump, an SPD-M20A Prominence diode-array detector, and a CBM-20A communications bus module. All the analytical HPLC analyses were performed using a Phenomenex (00G-4252-B0) Luna column (5 µm, C18 (2), 100 Å, 250 × 4.6 mm) with a flow rate of 0.5 mL/min at an ambient temperature and an injection volume of 10 µL. The solvents used were methanol-acetonitrile (4:3) containing 0.1% formic acid (solvent B) and H_2_O containing 0.1% formic acid (solvent A)**.** A gradient elution starting with a linear gradient from 40% B to 100% B in 28 min followed by maintaining at 100% B for the next 5 min then returning to 40% B in 2 min was used. Before the injection, the sample solution (1 mg/mL) was filtered using a PVDF membrane with a pore size of 0.45 µm.

### 3.2. Plant Material and Preparation of the Extract

The leaves of *V. xanthophloea* were collected from the University of KwaZulu-Natal (UKZN) botanical gardens, Pietermaritzburg Campus, in April 2017. The plant was identified by Alison Young, the curator of the botanical gardens. A voucher specimen with an accession number (*V. xanthophloea* NU0048529) was prepared and deposited at the Bews herbarium, UKZN School of Life Sciences. The plant material was air-dried at room temperature in the laboratory and crushed to a coarse powder using a hammer mill. The powdered material was weighed and stored in paper bags in an aerated environment.

For the antiplasmodial bioassay, the plant material (50 g) was extracted by cold maceration with constant stirring in 500 mL of dichloromethane-methanol (1:1, *v*/*v*) for 72 h. The extract was filtered and concentrated in vacuo using a rotary evaporator. *Acacia* species are notorious for their tannin content. Tannins bind non-selectively to most proteins and then give false-positive results in bioassays. Therefore, the fractionation of *V. xanthophloea* crude extracts was optimised to reduce or eliminate the tannin content as described by Wall et al. [44] with a slight modification. Briefly, the crude extract of *V. xanthophloea* (4.0 g) was reconstituted in 90% methanol and partitioned with hexanes (600 mL) to give a hexane fraction (0.8 g). The residual methanol extract was concentrated under a reduced pressure, redissolved in distilled water, and exhaustively partitioned with ethyl acetate (900 mL). The obtained ethyl acetate fraction was concentrated to a third of the initial volume and then partitioned with a 2% NaCl solution (200 mL × 4) to remove the tannins present. Finally, the residual ethyl acetate layer was dried over anhydrous MgSO_4_ and evaporated to produce a greenish-brown solid mass (1.3 g).

### 3.3. Isolation of the Compounds from the V. xanthophloea Leaf

The leaves of *V. xanthophloea* (1000 g) were extracted as described in Section 3.2. The resulting detanninised ethyl acetate fraction (4.5 g) was subjected to silica gel column chromatography and eluted with dichloromethane or combinations of dichloromethane-ethyl acetate (1:9, 2:8, and 3:7) and ethyl acetate to afford five fractions, A–E. Fraction A (147 mg), which had an intense orange colour, was purified on a silica gel column elution with a hexane-ethyl acetate gradient (9:1, 7:3, 1:1) to give four subfractions. The purification of subfraction 1 gave *E*-lutein (**13**) (3.3 mg). The purification of subfraction 2 by an isocratic elution with hexane-ethyl acetate (8:2) gave a mixture (4.0 mg) of 3,7,8,2′,4′,5′-hexamethoxyflavone (**1**) and 2′-hydroxy-3,7,8,4′5′-pentamethoxyflavone (**2**). Attempts to separate the two compounds using silica column chromatography, Sephadex LH-20, and TLC were not successful. Subfraction 3 was chromatographed on Sephadex LH-20 with an elution of methanol to give pinoresinol (**12**) (1.6 mg) and 5,7,2′-trihydroxy-3,4′,5′-trimethoxyflavone (**3**) (0.9 mg). Repeated Sephadex LH-20 chromatographic purifications of Fraction B (19 mg) by eluting with methanol gave kaempferol (**10**) (3 mg) and impure apigenin (**11**) (6 mg), which was further purified on a silica gel column by an isocratic elution with hexane-ethyl acetate (7:3). Silica gel column chromatography of Fraction C (52 mg) by an isocratic elution with hexane-ethyl acetate (9:1) gave 1-heptacosanol (**14**) (6 mg). Fraction D (340 mg) was initially purified on a Sephadex LH-20 column (eluted with methanol) and three subfractions were obtained from this process. Subfraction 1 was further chromatographed on a silica gel column to afford quercetin (**5**) (6 mg). Subfractions 2 and 3 were pooled together based on the similarity of the TLC profiles and repeatedly subjected to Sephadex LH-20 chromatography (methanol) to afford methyl gallate (**9**) (154 mg) and 3-*O*-methylquercetin (**4**) (9 mg). Repeated Sephadex LH-20 chromatography of Fraction E eluted with methanol gave taxifolin (**6**) (7.8 mg), catechin (**7**) (48.2 mg), and a mixture of catechin and gallocatechin (**8**) (6.5 mg).

The hexane fraction obtained from the partitioning of the *V. xanthophloea* crude extract was also purified by silica gel column chromatography with 100% hexanes and hexane-ethyl acetate combinations (9:1 and 8:2), affording a mixture of phytol (**15**) and lupeol (**16**).

Spectroscopic data of compounds:

3,7,8,2′,4′,5′-hexamethoxyflavone (**1**): pale yellow solid, UV (MeOH/ACN): λ_max_ 229, 331 nm; for ^1^H and ^13^C NMR data, see Table 1; HPLC R_t_: 25.919 min; HR-ESI-(+)-MS: *m/z* 403.1401 [M + H]^+^ (calculated for C_21_H_23_O_8_, 403.1393).

5,7,2′-trihydroxy-3,4′,5′-trimethoxyflavone (**3**): yellow solid, UV (MeOH/ACN): λ_max_ 256, 342 nm; for ^1^H and 13C NMR data, see Table 1; HPLC R_t_: 26.040 min; HR-ESI-(-)-MS: *m/z* 359.0776 [M − H]^−^ (calculated for C_18_H_15_O_8_, 359.0767).

### 3.4. Antimalarial Assay

#### 3.4.1. The Parasites

Malaria parasites (*Plasmodium falciparum* 3D7 clone) were maintained in an RPMI 1640 medium containing 2 mM L-glutamine and 25 mM Hepes (Lonza). The medium was further supplemented with 5% Albumax II, 20 mM glucose, 0.65 mM hypoxanthine, 60 µg/mL gentamycin, and 2–4% haematocrit human red blood cells. The parasites were cultured at 37 °C under an atmosphere of 5% CO_2_, 5% O_2_, and 90% N_2_ in a sealed T75 culture flask [45].

#### 3.4.2. Assessment of the In Vitro Antiplasmodial Activity

For the antiplasmodial activity, the extracts and isolated compounds were first tested at 50 µg/mL and 10 µg/mL in duplicate experiments. IC_50_ values determined from the dose-response curves were evaluated for the compounds only showing < 50% viability without cytotoxicity at 10 µg/mL. For the IC_50_ screening, a range of concentrations of the extracts and compounds starting from 100 µg/mL and extended by 3-fold serial dilution were tested in triplicate and standard deviations (SD) were derived where applicable. The parasite viability was determined by measuring the activity of parasite lactate dehydrogenase (pLDH) as described by Makler et al. [46]. Serial dilutions of the extracts and compounds were added to the in vitro cultures of *P. falciparum* (3D7 clone) in 96-well plates (1% haematocrit, 2% parasitaemia) and incubated for 48 h. To 125 µL solutions containing a mixture of Malstat (0.18 M lactic acid, 0.13 mM 3-acetylpyridine adenine dinucleotide, 0.16% Triton X-100, and 44 mM Tris, pH 9) and NBT/PES (0.39 mM nitro blue tetrazolium and 0.05 mM phenazine ethosulfate), 20 µL of the incubated culture was then added in a fresh 96-well plate. The absorbance reading of the purple product was formed, which was an indication of the pLDH activity and, hence, the number of parasites present was quantified in a Spectramax M3 microplate reader (Abs_620_). The % parasite viability, as indicated by the pLDH activity in the treated wells compared with the untreated controls, was calculated from the absorbance values for each compound/extract. The parasitised RBCs in the absence of test compounds/drug were taken as the untreated control with 100% viability. For each compound, the percentage viability was plotted against the log of the compound concentration and the IC_50_ value (50% inhibitory concentration) was obtained from the resulting dose-response curve by a non-linear regression. Chloroquine was used as a positive control drug (IC_50_ values ranged from 0.01–0.05 µM).

#### 3.4.3. In Vitro Cytotoxicity Assay

The cytotoxicity of the compounds/extracts against HeLa (human cervix adenocarcinoma) cells (Cellonex, South Africa) was evaluated by plating the cells at a density of 2 × 10^4^ cells per well in a DMEM medium supplemented with 10% foetal bovine serum and penicillin/streptomycin/amphotericin B antibiotics in 96-well plates. After incubating for 24 h at 37 °C in a humidified 5% CO_2_ incubator, the test samples were added at a final concentration of 50 µg/mL or 10 µg/mL to the cells in triplicate wells (final volume per well was 200 µL) and further incubated or 24 h. Subsequently, a 20 µL resazurin stock solution (0.6 mM resazurin in phosphate-buffered saline) was added to each well and, after a 4 h incubation, the fluorescence (Exc_560_/Emm_590_) was measured on a Spectramax M3 plate reader. The percentage cell viability in the treated wells relative to the wells containing the untreated control cells was calculated from the fluorescence values after subtracting the background readings obtained from the empty wells. Emetine was used as the positive control.

## 4. Conclusions

The chemical investigation of *V*. *xanthophloea* leaves resulted in the isolation of 2 new flavonoids and 14 other known compounds. To the best of our knowledge, this is the first report on the chemical constituents of *V*. *xanthophloea* despite its widespread use. Methyl gallate was responsible for the antiplasmodial activity observed for the plant extract.

## Figures and Tables

**Figure 1 pharmaceuticals-15-00470-f001:**
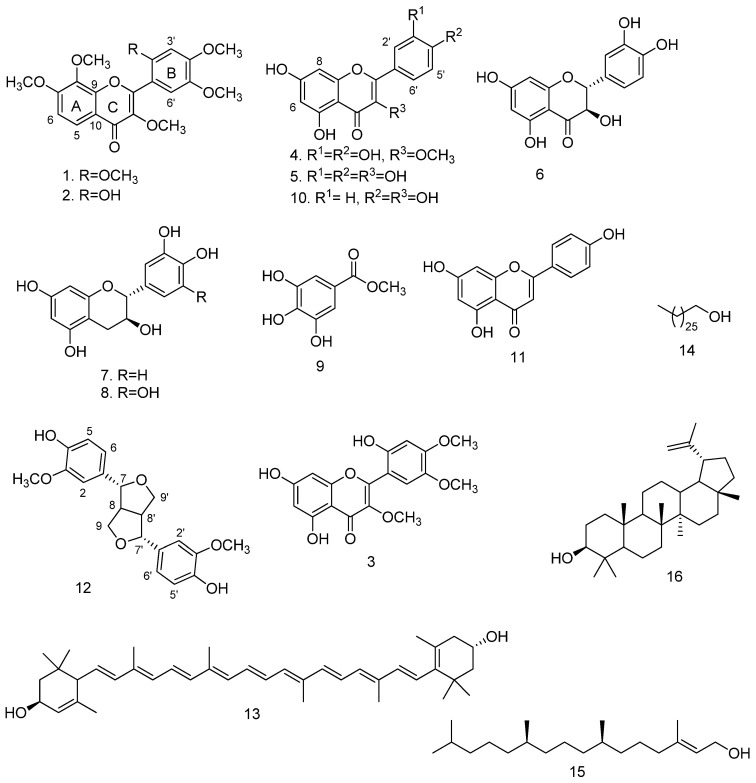
Structures of the compounds isolated from *V. xanthophloea*.

**Table 1 pharmaceuticals-15-00470-t001:** ^1^H and ^13^C NMR data of compounds **1** and **3**.

Position	1		3	
	δ_H_	δ_C_	δ_H_	δ_C_
2		155.7		156.6
3		141.3		138.6
4		173.4		180.3
5	8.00 (d, *J* = 9.0 Hz)	121.0		161.5
6	7.03 (d, *J* = 9.0 Hz)	109.8	6.20 (d, *J* = 2.0 Hz)	99.7
7		156.1		166.2
8		136.8	6.33 (d, *J* = 2.0 Hz)	94.7
9		149.9		158.2
10		119.6		105.8
1′		111.4		109.4
2′		151.9		151.0
3′	6.64, (s)	97.7	6.60, (s)	101.5
4′		152.7		153.4
5′		143.0		142.5
6′	7.32, (s)	110.9	7.02, (s)	113.0
OCH_3_-3	3.80, (s)	60.4	3.73, (s)	60.7
OCH_3_-7	3.98, (s)	56.6		
OCH_3_-8	3.95, (s)	61.5		
OCH_3_-2′	3.97, (s)	56.3		
OCH_3_-4′	3.85, (s)	56.7	3.88, (s)	55.4
OCH_3_-5′	3.86, (s)	56.8	3.83, (s)	56.4

**Table 2 pharmaceuticals-15-00470-t002:** In vitro antiplasmodial activity of the extract and a few isolated compounds.

Compound	IC_50_ (µg/mL)	Viability % ± SD at 50 μg/mL	Viability % ± SD at 10 μg/mL
*V. xanthophloea*	10.6	-	-
Methyl gallate (**9**)	1.2 ± 0.07 (6.52 µM)	17.7 ± 1.5	26.9 ± 0.7
3-*O*-Methylquercetin (**4**)	Nd	21.9 ± 1.5	82.9 ± 3.0
Mixture (1:1) of 3,7,8,2′,4′,5′-hexamethoxyflavone (**1**) and 2′-hydroxy-3,7,8,4′,5′-pentamethoxyflavone (**2**)	Nd	14.1 ± 0.7	73.0 ± 0.7
Kaempferol (**10**)	25.0 (87.3 µM)	-	-
Dihydroquercetin (**6**)	27.6 (90.71 µM)	-	-
Mixture (1:1) of 3-*O*-methylquercetin (**4**) and methyl gallate (**9**)	4.6	-	-
Chloroquine	0.014 µM	-	-

Nd = not done.

**Table 3 pharmaceuticals-15-00470-t003:** In vitro cytotoxic activity of the extract and a few isolated compounds against HeLa cells.

Compound	Viability % ± SD at 50 μg/mL	Viability % ± SD at 10 μg/mL
*V. xanthophloea*	* 41%	# 98%
Methyl gallate (**9**)	2.0 ± 0.2	68.6 ± 2.0
3-*O*-Methylquercetin (**4**)	4.5 ± 0.1	57.9 ± 5.2
Mixture (1:1) of 3,7,8,2′,4′,5′-hexamethoxyflavone (**1**) and 2′-hydroxy-3,7,8,4′,5′-pentamethoxyflavone (**2**)	7.1 ± 0.4	63.1 ± 4.3
Emetine	IC_50_ = 0.033 µM	IC_50_ = 0.04 µM

# = % cell viability at 33 µg/mL; * = % cell viability at 100 µg/mL. Emetine is known to be cytotoxic to HeLa cells.

## Data Availability

Data is contained within the article and Appendix A.

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
