# Peer review of "Antiplasmodial Activity of Vachellia xanthophloea (Benth.) P.J.H. Hurter (African Fever Tree) and Its Constituents"

_pharmaceuticals, 2022, doi:10.3390/ph15040470_

Round 1

Reviewer 1 Report

The manuscript presents the results of a study to determine the structures of compounds present in the leaves of Vachellia xanthophloea and the results of tests for their antiplasmodial activity. A very detailed description of the structures of the two new flavonoids is to be appreciated, especially since the compounds were obtained as an equimolar mixture.

However, the second part concerning biological studies lacks some information.

Minor remarks

  1. Line 89 gives the wrong molecular formula, it should be C21H22O8.
  2. The same spectral data for compounds 1 and 3 were included twice in the manuscript. The 1H and 13C NMR data of compounds 1 and 3 were included in Table 1 (only the H-3' derived signal information for compound 1 is missing). Exactly the same data, supplemented with additional information was found in Section 3.3 as "Spectroscopic data of compounds" (Line 306 -323). In this section, when describing the 1H NMR spectra, the location of the CH3 groups is not given. I suggest leaving only one of these versions (appropriately supplemented) in the final manuscript.
  3. According to the text, Table 2 and 3 present the results of the biological studies conducted on the activity of the extract and the compounds isolated from it. In contrast, only selected results are presented in the Tables. Why?
  4. Table 2 - compound 9 was tested, compound 4 was not tested. Then where did the idea to test compound 4 and 9 come from?

Author Response

Point-by-point response to the reviewers’ comments

Reviewer 1

Comments and Suggestions for Authors

We thank the reviewer for the positive comment and valuable suggestions.

Minor remarks

  1. Line 89 gives the wrong molecular formula, it should be C21H22O8.

The mistake in the molecular formula has been corrected to C21H22O8 as rightly observed by the reviewer. We thank the reviewer for diligence.

  1. The same spectral data for compounds 1 and 3 were included twice in the manuscript. The 1H and 13C NMR data of compounds 1 and 3 were included in Table 1 (only the H-3' derived signal information for compound 1 is missing). Exactly the same data, supplemented with additional information was found in Section 3.3 as "Spectroscopic data of compounds" (Line 306 -323). In this section, when describing the 1H NMR spectra, the location of the CH3 groups is not given. I suggest leaving only one of these versions (appropriately supplemented) in the final manuscript.

Again, we thank the reviewer for thoroughness and for the suggestion. We have removed the 1H and 13C NMR data from section 3.3 and referred the reader to Table 1 for this information. Also, the missing chemical shift value for H-3' in Table 1 has been included. In the process of correcting this mistake, we noticed the chemical shifts of H-3' and H-6' for compound 3 should be swapped and this has also been corrected

  1. According to the text, Table 2 and 3 present the results of the biological studies conducted on the activity of the extract and the compounds isolated from it. In contrast, only selected results are presented in the Tables. Why?

The bioassay part of the work was done by our collaboration partners who could only test a limited number of compounds, so we had to prioritize which compounds were selected. For this reason, compounds like lupeol, lutein, heptacosanol, and phytol, that are too lipophilic and/or ubiquitous plant metabolites were given a low priority. Also some of the compounds, pinoresinol and gallocatechin were isolated in insufficient amount so we could not test them for bioactivity, besides these two compounds have been previously reported to exert no antiplasmodial activity. We have included this information in the manuscript with appropriate citations. To show that not all the compounds were tested, we have added the phrase “… some of the…” at the beginning of section 2.2.  

  1. Table 2 - compound 9 was tested, compound 4 was not tested. Then where did the idea to test compound 4 and 9 come from?

Compound 4 was actually tested alone, first at 50 μg/mL and then at 10 μg/mL, it showed good activity at the higher concentration (50 μg/mL) but the activity was diminished at the lower concentration (10 μg/mL). Therefore its IC50 was not determined because we had a set a strict criteria that only compounds showing > 50% inhibition at 10 μg/mL (which we consider not to be an unrealistic high concentration for an in vitro cell based assay) could be described as active and worthy of a dose response curve to determine IC50.

Reviewer 2 Report

The authors isolated, purified, and characterized the compounds extracted from the leaves of Vachellia xanthophloea, reputed to possess antimalarial properties in African traditional medicine, and tested their antimalarial and cytotoxic activities. The ethyl acetate fraction yielded 2 new flavonoids and 14 known compounds. Of these components in the crude extract, methyl gallate was identified to be the compound with antimalarial activity (IC50 1.2 µg/mL).

The background section is adequate and very informative for uninitiated readers. The methods are described clearly in detail. The in vitro assay to determine antimalarial activity (based on colorimetric reading) is no longer used in many specialized laboratories but is adequate for drug screening. The results are presented in a logical and clear manner. Tables and figures, as well as supplementary data, are well presented and are very helpful for the understanding of the authors’ work. Discussion is clear and pertinent and includes limitations of the study. The paper is generally well written.

The abstract of the work was published earlier by MDPI in 7th Int Electronic Conference on Med Chem. A similar work on flavonoids from Pappea capensis was also published by the same authors in MDPI (Molecules 2021;26:3875, doi:10.3390/molecules26133875). The wordings of the present manuscript in lines 206–209, lines 227–240, lines 258–260, lines 333–355, and lines 357–367 are very similar or identical to those in Tajuddeen et al. (Molecules, 2021;26:3875). The text in these lines should be revised to avoid “self-plagiarism.”

I suggest the authors to explain what the abbreviations stand for when first used in the text and/or provide a list of abbreviations at the end of the main text. General readers may not necessarily understand all of the abbreviations used in the text.

Major comments:

none

Minor comments:

Line 70. “NMR” abbreviation

Line 77. “HR-ESI-MS” abbreviation

Line 85. “DEPT NMR” abbreviation

Line 87. “HSQC” abbreviation

Line 94. “HMBC spectrum” abbreviation

Line 99. “COSY correlation” abbreviation

Lines 157-162. This long sentence (“Several phenolic acid derivatives…from A. confuse Merr [35]”) can be divided into three or four shorter sentences.

Line 163. “Lignans are not commonly found in the Vachellia (delete the comma) and Senegalia genera [36]. (place a period here and start a new sentence) In fact, the isolation…”

Lines 183-184. “when a mixture…was tested…”

Lines 191-192. “most of the flavonoids were also cytotoxic to HeLa cells…”

Line 231. “TOF” abbreviation, time of flight

Line 232. “ESI ionization” abbreviation

Line 233. “GC-MS” abbreviation

Line 234-235. “For thin-layer chromatographic (TLC) analyses, pre-coated TLC silica gel…”

Line 243. “HPLC” abbreviation, high-performance liquid chromatography

Line 252. “PVDF” abbreviation

Line 326. P. falciparum 3D7 is a clone (not just a “strain”) in most laboratories in the world.

Line 327. “HEPES” abbreviation

Lines 380-382. Delete “Please add.” “ ; delete also “and” after the grant number. The quotation marks should be deleted also.

Table 3. It would be helpful for readers if the authors can add a sentence, either in the table legend or in the methods section (section 3.4.3), explaining why emetine was included in cytotoxicity tests.

REF 16. The online ahead of print (in 2021) may have been published. If so, please provide the complete reference (vol, pages).

REF 33. Article title, Acacia nilotica (small letter “n” in nilotica)

REF 38 is not complete. PLoS One 2013, 8, e79544. then doi number.

REF 41. Malar J. 2011, 10 (Suppl 1), S1

REF 44. Article title, Plasmodium falciparum

Author Response

Point-by-point response to the reviewers’ comments

Reviewer 2

Comments and Suggestions for Authors

We are grateful to the reviewer for the positive comments, taking the time to read our manuscript, and for the insightful suggestions.

The abstract of the work was published earlier by MDPI in 7th Int Electronic Conference on Med Chem. A similar work on flavonoids from Pappea capensis was also published by the same authors in MDPI (Molecules 2021;26:3875, doi:10.3390/molecules26133875). The wordings of the present manuscript in lines 206–209, lines 227–240, lines 258–260, lines 333–355, and lines 357–367 are very similar or identical to those in Tajuddeen et al. (Molecules, 2021;26:3875). The text in these lines should be revised to avoid “self-plagiarism.”

The sentences have been changed as follows,

Lines 206–208: Nevertheless, the further development of flavonoids as viable antiplasdmodial agents has been hampered by the apparent lack of selectivity in their bioactivity. Please see the updated manuscript

Lines 227–240: This part of the general procedures has been paraphrased as recommended by the reviewer. Please see the updated manuscript

Lines 258–260: A voucher specimen with accession number (V. xanthophloea NU0048529) has been prepared and deposited at the Bews herbarium, UKZN School of Life Sciences. Please see the updated manuscript

Lines 333–355: The text describing the antiplasmodial activity has been changed as recommended by the reviewer. Please see the updated manuscript

Lines 357–367: The procedure for the cytotoxicity assay has been rewritten as suggested by the reviewer. Please see the updated manuscript

I suggest the authors to explain what the abbreviations stand for when first used in the text and/or provide a list of abbreviations at the end of the main text. General readers may not necessarily understand all of the abbreviations used in the text.

We have explained the meaning of the abbreviations where they are first mentioned and provided a list of abbreviation at the end of the main text after the conclusion section

Minor comments:

Line 70. “NMR” abbreviation

Full meaning has been added where it was first mentioned

Line 77. “HR-ESI-MS” abbreviation

Full meaning has been added where it was first mentioned

Line 85. “DEPT NMR” abbreviation

Full meaning has been added where it was first mentioned

Line 87. “HSQC” abbreviation

Full meaning has been added where it was first mentioned

Line 94. “HMBC spectrum” abbreviation

Full meaning has been added where it was first mentioned

Line 99. “COSY correlation” abbreviation

Full meaning has been added in the list of abbreviations

Lines 157-162. This long sentence (“Several phenolic acid derivatives…from A. confuse Merr [35]”) can be divided into three or four shorter sentences.

The long sentence has been divided into three shorter ones. Please see the updated manuscript

Line 163. “Lignans are not commonly found in the Vachellia (delete the comma) and Senegalia genera [36]. (place a period here and start a new sentence) In fact, the isolation…”

We have added a full stop as suggested by the reviewer and started a new sentence from “In fact…”

Lines 183-184. “when a mixture…was tested…”

The mistake has been corrected by changing were to was

Lines 191-192. “most of the flavonoids were also cytotoxic to HeLa cells…”

The mistake has been corrected

Line 231. “TOF” abbreviation, time of flight

Full meaning has been added where it was first mentioned

Line 232. “ESI ionization” abbreviation

Full meaning has been added where it was first mentioned

Line 233. “GC-MS” abbreviation

Full meaning has been added in the list of abbreviation

Line 234-235. “For thin-layer chromatographic (TLC) analyses, pre-coated TLC silica gel…”

Full meaning has been added in the list of abbreviation

Line 243. “HPLC” abbreviation, high-performance liquid chromatography

Full meaning has been added at first mention and in the list of abbreviation

Line 252. “PVDF” abbreviation

Full meaning has been added in the list of abbreviation

Line 326. P. falciparum 3D7 is a clone (not just a “strain”) in most laboratories in the world.

Strain has been replaced by clone for 3D7 P. falciparum throughout in the manuscript

Line 327. “HEPES” abbreviation

Full meaning has been added in the list of abbreviation

Lines 380-382. Delete “Please add.” “ ; delete also “and” after the grant number. The quotation marks should be deleted also.

“please add.”, quotations and colon have been deleted

Table 3. It would be helpful for readers if the authors can add a sentence, either in the table legend or in the methods section (section 3.4.3), explaining why emetine was included in cytotoxicity tests.

We have added a statement to show that emetine was used as the positive control at the end of section 3.4.3 and as a footnote in Table 3

REF 16. The online ahead of print (in 2021) may have been published. If so, please provide the complete reference (vol, pages).

The full bibliographic information for this article is still not available because the article has still not been published in an issue of the journal

REF 33. Article title, Acacia nilotica (small letter “n” in nilotica)

The “N” in Nilotica has been changed to small letter “n” nilotica

REF 38 is not complete. PLoS One 2013, 8, e79544. then doi number.

The missing information, i.e. e79544, in reference 38 has been added

REF 41. Malar J. 2011, 10 (Suppl 1), S1

The missing information in reference 41, i.e. (Suppl 1), S1, has been added

REF 44. Article title, Plasmodium falciparum

The title has been corrected by removing - in Plasmodium falciparum and making “p” uppercase